# Dysregulation of Peripheral Blood Mononuclear Cells and Immune-Related Proteins during the Early Post-Operative Immune Response in Ovarian Cancer Patients

**DOI:** 10.3390/cancers16010190

**Published:** 2023-12-30

**Authors:** Jonas Ulevicius, Aldona Jasukaitiene, Arenida Bartkeviciene, Zilvinas Dambrauskas, Antanas Gulbinas, Daiva Urboniene, Saulius Paskauskas

**Affiliations:** 1Laboratory of Surgical Gastroenterology, Institute for Digestive Research, Medical Academy, Lithuanian University of Health Sciences, A. Mickeviciaus g. 9, LT-44307 Kaunas, Lithuania; aldona.jasukaitiene@lsmu.lt (A.J.); arenida.bartkeviciene@lsmu.lt (A.B.); zilvinas.dambrauskas@lsmu.lt (Z.D.); antanas.gulbinas@lsmu.lt (A.G.); 2Department of Laboratory Medicine, Medical Academy, Lithuanian University of Health Sciences, A. Mickeviciaus g. 9, LT-44307 Kaunas, Lithuania; daiva.urboniene@lsmu.lt; 3Department of Obstetrics and Gynecology, Medical Academy, Lithuanian University of Health Sciences, A. Mickeviciaus g. 9, LT-44307 Kaunas, Lithuania; saulius.paskauskas@lsmu.lt

**Keywords:** ovarian cancer, surgery, PBMC, IL-1β, IL-4, IL-6, IL-10, PD-1, PD-L1, HO-1

## Abstract

**Simple Summary:**

This study investigated the immune response in ovarian cancer (OC) patients before and after surgery. The aim was to explore how the immune system is affected by surgery, with a specific focus on peripheral blood mononuclear cells (PBMCs). An analysis of blood samples revealed pre-operative immune imbalances in OC patients that were further exacerbated post-operatively. The study results suggest that OC patients experience a degree of immune suppression, particularly during the early post-operative period. This indicates a potential window of vulnerability that could facilitate cancer progression. Our findings may contribute to the development of better treatment strategies and improved outcomes for OC patients.

**Abstract:**

Surgical treatment is a cornerstone of ovarian cancer (OC) therapy and exerts a substantial influence on the immune system. Immune responses also play a pivotal and intricate role in OC progression. The aim of this study was to investigate the dynamics of immune-related protein expression and the activity of peripheral blood mononuclear cells (PBMCs) in OC patients, both before surgery and during the early postoperative phase. The study cohort comprised 23 OC patients and 20 non-cancer controls. A comprehensive analysis of PBMCs revealed significant pre-operative downregulation in the mRNA expression of multiple immune-related proteins, including interleukins, PD-1, PD-L1, and HO-1. This was followed by further dysregulation during the first 5 post-operative days. Although most serum interleukin concentrations showed only minor changes, a distinct increase in IL-6 and HO-1 levels was observed post-operatively. Reduced metabolic and phagocytic activity and increased production of reactive oxygen species (ROS) were observed on day 1 post-surgery. These findings suggest a shift towards immune tolerance during the early post-operative phase of OC, potentially creating a window for treatment. Further research into post-operative PBMC activity could lead to the development of new or improved treatment strategies for OC.

## 1. Introduction

Ovarian cancer (OC) is the eighth most common cancer type in women worldwide, resulting in over 200,000 deaths annually. Most OC patients are diagnosed with advanced-stage disease, making it more difficult to treat [1,2]. Surgical removal of visible cancer tissue and chemotherapy are the most commonly employed treatment methods for OC. When combined appropriately, these two treatment approaches can significantly increase the rates of progression-free survival and overall survival [3]. Chemotherapy is rarely administered as a stand-alone treatment in OC, with surgery being a crucial independent prognostic factor [3,4,5]. Nevertheless, even with the most effective surgical strategies and chemotherapy protocols, disease relapse is frequent, and long-term remission remains a challenge [6]. Consequently, both established and novel approaches are continually being investigated in the quest to improve treatment strategies [7,8].

Despite its benefits for the treatment of OC, surgery can cause tissue trauma and exert systemic effects on the patient [9,10]. It significantly influences the immune response through various mechanisms, including the release of damage-associated molecular patterns, generation of neutrophil extracellular traps, and the activation of myeloid-derived suppressor cells, regulatory T cells, and programmed death ligand-1 [9]. Considering the well-documented interplay between the immune system, the tumour microenvironment, and the progression of OC [6], it is reasonable to hypothesise that surgery could also impact the progression of OC. Although the precise effect on patients is not yet fully understood, surgical trauma has been shown to disrupt immune function, thereby facilitating the formation of metastases [9,11,12]. This phenomenon has been demonstrated in patients undergoing surgical treatment for colorectal, pancreatic, breast, and lung cancers, as well as various other cancer types [13,14,15]. However, specific data for OC patients are currently lacking.

Peripheral blood mononuclear cells (PBMCs) constitute a cohort of immune cells that are pivotal for the generation of anticancer immunity. After settling in the tumour tissue, PBMCs interact with cancer cells and affect cancer progression [6,16,17]. Despite extensive data on the tumour microenvironment [6,18] and the serum cytokine profile of OC patients [19,20], little is known regarding cytokine production by PBMCs [21,22]. Furthermore, there have been few investigations of post-surgical immune responses in OC patients [23,24,25]. Consequently, there are almost no published data on the activity of PBMCs in OC patients following surgical intervention.

To gain a better understanding of the relationship between OC progression and surgical treatment, it is crucial to investigate how PBMCs and immune-related proteins [6,26,27,28] respond to surgical interventions in OC patients. Investigating interleukins IL-1β, IL-4, IL-6, and IL-10 is fundamental for comprehending the immune intricacies in anticancer immunity. These interleukins play crucial roles in regulating inflammatory responses (IL-1β and IL-6), modulating immune cell function and differentiation (IL-4 and IL-10), and maintaining the delicate balance between pro- and anti-inflammatory signals [6,27]. The interaction between immune cells and cancer extends beyond cytokine production. Immune checkpoints like programmed cell death protein 1 (PD-1) and programmed cell death protein ligand 1 (PD-L1) promote immune tolerance and, when activated, can facilitate tumour progression [29,30]. Heme oxygenase 1 (HO-1), a well-known enzyme involved in eliminating oxidative stress, predominantly exhibits anti-inflammatory and immune-suppressive properties when expressed by tumour microenvironment cells [26]. While data exist showing the profiles of PD-1/PD-L1 and HO-1 in the serum and tumour microenvironment [29,31], we lack information on their expression in PBMCs post-operatively in OC patients.

We hypothesised that in the early post-operative period, OC patients experience an imbalance in their immune system that is closely associated with the function of PBMCs. The aim of the current study was to examine potential alterations in PBMC activity, concentration of interleukins, and other immune-related proteins in surgically treated OC patients, with a particular focus on the early post-operative period. By drawing parallels with findings from other cancer types, this work could potentially improve the treatment strategies for OC.

## 2. Materials and Methods

### 2.1. Study Design and Patient Selection

This prospective study involved a cohort of 23 patients diagnosed with epithelial OC and surgically treated between January 2021 and April 2023 at the Lithuanian University of Health Sciences Hospital, Kaunas clinics. Laparotomy and cytoreduction were performed for OC patients who had not undergone prior chemotherapy. The selection criteria for the study group were patients diagnosed with primary OC and FIGO stage III and IV [32]. The exclusion criteria included individuals with autoimmune disease, other confirmed or suspected cancer, and patients who had undergone surgery or received blood transfusions within the past month. Histopathological examination confirmed the diagnosis of OC. The control group consisted of 20 women who were age- and body mass index (BMI)-matched with the OC patients and were cancer-free. The same exclusion criteria used for the study group were applied to the controls. All participants gave informed consent before inclusion in the study. This research study adhered to the principles outlined in the Declaration of Helsinki and was approved by the Kaunas Regional Biomedical Research Ethics Committee (No. BE-2-16).

### 2.2. Blood Collection

Samples of peripheral venous blood were collected from OC patients and healthy controls. For OC patients, the blood samples were obtained prior to surgery, and at 1, 3, and 5 days after surgery. After reviewing the pilot results, we observed a consistent shift in the data between days 1 and 5 after the surgery. Consequently, certain assays were excluded from day 3 protocol. Blood sampling and the assay schedule for OC patients are outlined in Figure 1. Blood sampling from healthy controls was a one-time procedure for each participant and followed the same assay protocol used for the preoperative assessment of OC patients. Each blood collection consisted of three 10 mL vacutainers containing EDTA K2 (BD, Plymouth, UK), and two 3.5 mL vacutainers with serum clot activator (SST) from Weihai Hongyu (Weihai, China).

### 2.3. Isolation of PBMCs and Serum Preparation

Following collection in vacutainers with EDTA K2, blood samples were centrifuged for 10 min at 2470× *g* for plasma separation. PBMCs were isolated by gradient centrifugation with Ficoll-Paque PREMIUM medium (Cytiva, Uppsala, Sweden), according to the manufacturer’s protocol [33]. They were then carefully collected by pipetting, and washed twice with PBS (Sigma-Aldrich, Taufkirchen, Germany) at 250× *g* for 5 min each. For downstream experiments, separate aliquots of the PBMCs were used for cell culture (PBMC functional evaluation by fluorometric or spectrophotometric assays) and Western blotting (WB). An aliquot was also mixed with RNAlater^®^ (Thermo Fisher Scientific, Waltham, MA, USA) and stored at −20 °C or −80 °C for use in subsequent assays.

Blood samples collected in vacutainers with SST were allowed to sit for an hour to facilitate clot formation. Vacutainers were then centrifugated at 2500× *g* for 20 min at 4 °C. The resultant serum, situated above the gel layer, was meticulously retrieved and transferred into cryogenic tubes for storage at −80 °C until analysis.

### 2.4. RNA Extraction and Real-Time Polymerase Chain Reaction (RT-PCR)

Total RNA extraction from pre-prepared PBMC samples was performed using an RNA extraction kit (Abbexa, Cambridge, UK), as per the manufacturer’s instructions [34]. The amount of purified RNA was quantified, and its purity was assessed with UV spectrophotometry (NanoDrop 2000, Thermo Fisher Scientific, Waltham, MA, USA). cDNA was synthesised from 2 μg of RNA using the High/Medium-Capacity cDNA Reverse Transcription Kit (Applied Biosystems, Waltham, MA, USA) and the 7500 Fast Real-Time PCR System (Applied Biosystems, Waltham, MA, USA). Amplification of specific RNA was performed in a 20 μL reaction mixture containing 2 μL of template cDNA, 1 μL of primers, 10 μL of TaqMan™ Universal PCR Master Mix (Applied Biosystems, Waltham, MA, USA), and 7 μL of nuclease-free water. The following primers (Applied Biosystems, Waltham, MA, USA) were used: IL-1β (Hs001555410), IL-4 (Hs00174122), IL-6 (Hs00174131), IL-10 (Hs00961619), HO-1 (Hs0111025), PD-1 (Hs05043241), PD-L1 (Hs00204257), and GAPDH (Hs02786624). Primer efficacy and melting curve analyses were performed to ensure specific amplification. All experiments were conducted in triplicate and repeated as needed for result validation.

### 2.5. Western Blot Analysis

Proteins were extracted from PBMCs using RIPA lysis buffer (Abcam, Cambridge, MA, USA) supplemented with protease-phosphatase inhibitors (Roche, Basel, Switzerland). After centrifugation at 10,000× *g* for 10 min, the supernatants were collected and stored at −80 °C. The BCA protein assay kit (Thermo Fisher Scientific, Waltham, MA, USA) was used to measure protein concentration. Protein samples (45 µg) were mixed with Bolt MES SDS loading buffer (Invitrogen, Carlsbad, CA, USA) and heated at 97 °C for 5 min. The mixture was then loaded onto a 4–12% sodium dodecyl sulphate-polyacrylamide gel electrophoresis (SDS-PAGE) gel and subsequently transferred onto poly-vinylidene fluoride (PVDF) membranes for 40 min at 30 V. The membranes were then exposed to a 5% blocking buffer (Invitrogen, Waltham, MA, USA) for 40 min at room temperature. They were subsequently incubated with primary antibodies either overnight at 4 °C or for 1 h at room temperature. The following primary antibodies were used: rabbit polyclonal anti-HO-1 (ab137749, Abcam, Cambridge, MA, USA) at a 1:1000 dilution, and mouse monoclonal anti-GAPDH (AM4300, Invitrogen, Waltham, MA, USA) at a 1:3000 dilution. Following incubation with a primary antibody, the membranes were washed and then incubated with the appropriate horseradish-peroxidase-conjugated (HRP) or alkaline phosphatase-conjugated (AP) secondary antibody (HRP) from Invitrogen (Carlsbad, CA, USA). Incubation times were either 1 h or 30 min at 37 °C. Subsequent rounds of washing were performed before the membranes were exposed for 5 min to chemiluminescence substrates (Invitrogen, Carlsbad, CA, USA) or West Pico Stable peroxidase buffer with luminol enhancer (Thermo Scientific, Waltham, MA, USA). The protein bands were visualised with a ChemiDoc Imaging System (Bio-Rad Laboratories, Hercules, CA, USA). Subsequent quantitative analysis was performed using ImageJ software version 1.53a (National Institutes of Health, Bethesda, MD, USA).

### 2.6. PBMC Culture and Assessment of Activity

After isolation from patients and healthy participants, PBMCs were resuspended at a concentration of 1 million cells per mL for cell activity analysis. The samples were grown in an RPMI 1640 medium (Gibco Life Technologies Limited, Paisley, UK) without phenol red and supplemented with 10% foetal bovine serum (FBS) (Gibco Life Technologies Limited, Paisley, UK) and 1% penicillin/streptomycin (Gibco Life Technologies Limited, Paisley, UK). After incubating the samples for 30 min at 37 °C, they were distributed into dark and clear 96-well plates, with each well containing 90,000 cells suspended in the medium. Following 3 h of activation with various effectors, the dark well plates were used in fluorometric or spectrophotometric assays to evaluate the functional activity of PBMCs. Specifically, these were phagocytosis and the production of reactive oxygen species (ROS) and nitrous oxide (NO). Simultaneously with functional activity tests, PBMC viability was determined through resazurin metabolism evaluation. Clear well plates were used to assess metabolic activity with the AlamarBlue^®^ assay (Thermo Fisher Scientific, Waltham, MA, USA). Similar to the procedure for the functional assays (described below), three categories of wells were prepared: wells containing the specific activator, negative controls, or media controls. After cell activation, the AlamarBlue^®^ assay was performed 4 hours later in accordance with the manufacturer’s instructions. This was followed by spectrophotometric measurements (The Sunrise, Tecan, Grodig, Austria) [35]. For baseline viability assessment, resazurin metabolism was concurrently measured in wells containing PBMCs without the activators. Resazurin metabolism was subsequently evaluated through fluorometric analysis.

### 2.7. Assessment of Phagocytosis

Phagocytosis, the process by which cells engulf and eliminate foreign particles or pathogens, is a critical mechanism used by immune cells, particularly macrophages and dendritic cells. Assessing phagocytic activity serves as a valuable marker for immune cell function in the context of anticancer immunity due to its role in clearing cancer cells, debris, and promoting an anti-tumour immune response [30,36]. Lipopolysaccharide (LPS) was used to induce phagocytosis in PBMCs. LPS (Sigma-Aldrich, Rehovot, Israel) was introduced into three wells of a 96-well dark plate, with each well receiving 10 µL of a 5 µg/mL solution. Negative controls were established in another three wells using 10 µL of LPS diluent. The final three wells acted as media controls and contained only RPMI and 10 µL of 5 µg/mL LPS without cells. After 2 h of incubation and activation of phagocytic function, the dark plate was subjected to centrifugation and the supernatant was then removed. Subsequently, 100 µL of pHrodo Green Zymosan Bioparticles (Invitrogen, Eugene, OR, USA) [37] was added to each well, followed by incubation at 37 °C for one hour. Fluorescence measurements were performed using a fluorimeter (Fluoroskan Ascent, Thermo Fisher Scientific, Waltham, MA, USA), with excitation set at 510 nm and detection at 538 nm.

### 2.8. Assessment of ROS Production

ROS production is significant for evaluating immune cell function, given its role in diverse immunological processes such as signalling, inflammation, and pathogen defence. Particularly in anticancer immunity, ROS acts as a marker due to its involvement in mediating cytotoxic effects against tumours [30,38]. The production of ROS was evaluated using tert-butyl hydroperoxide (TBHP) as an organic peroxide and oxidative-stress-inducing agent. PBMCs were stained with DCFDA (Cellular ROS Assay Kit, Abcam, Cambridge, UK) and then treated with 10 µL of 2.5 mM TBHP (Abcam, Cambridge, UK), according to the manufacturer’s protocol [39]. TBHP was administered to three wells, while another set of three wells were designated as negative controls with 10 µL of TBHP diluent. Three wells received only RPMI with 10 µL of 2.5 mM TBHP, thus serving as media controls without cells. Following a 3 h incubation at 37 °C, fluorescence measurements were conducted using a fluorimeter (Fluoroskan Ascent, Thermo Fisher Scientific, Waltham, MA, USA) with excitation set at 485 nm and detection at 538 nm.

### 2.9. Assessment of NO Production

NO has a dual role in oxidative stress regulation, acting both as a free radical and in balancing oxidative stress levels within cells. It plays a significant role in immune responses against cancer cells, contributing to tumour cell destruction and regulating anti-tumour immune responses [30,40,41]. Nitrite production in PBMCs was induced using LPS and L-arginine [42]. NO production was assessed in clear 96-well plates after 3 h of treatment with 10 µL of LPS (Sigma-Aldrich, Rehovot, Israel) at a concentration of 5 µg/mL in conjunction with 100 mM of L-arginine (Sigma-Aldrich, Tokyo, Japan). LPS and L-arginine were introduced into three wells, while an additional three wells were designated as negative controls and contained 10 µL of LPS and L-arginine diluent. In three additional wells, RPMI with 10 µL of 5 µg/mL LPS and 100 mM of L-arginine was added without cells as a medium control. After 2.5 h of activation at 37 °C, the samples were combined with 100 μL of Griess reagent (Invitrogen, Eugene, OR, USA) [43] and incubated at room temperature for 30 min. This was followed by spectrophotometric measurement (The Sunrise, Tecan, Grodig, Austria) at 550 nm absorption.

### 2.10. Analysis of Serum Cytokines Using Luminex

Serum levels of free IL-1β, IL-4, IL-6, and IL-10 were measured using magnetic bead-based multiplex assays (Human Cytokine Premixed Multi-Analyte Kit from R&D) and a Luminex^®^ 100 analyser (Luminex Corporation, Austin, TX, USA). Frozen serum samples were thawed and then centrifuged for 4 min at 4 °C and 16,000× *g* to eliminate any debris or precipitates. The subsequent steps followed the manufacturer’s protocol. Analyte-specific antibodies were pre-coated onto magnetic microparticles containing embedded fluorophores, each set at specific ratios for individual microparticle regions. Microparticles, standards, and samples were dispensed into wells, leading to immobilisation of the antibodies on target substances. After washing to remove unbound components, the samples were exposed to a mixture of biotinylated detection antibodies and a streptavidin–phycoerythrin (SAPE) reporter. Further washes eliminated unbound SAPE, and the microparticles were then resuspended in a buffer for analysis with the Luminex^®^ 100 instrument. The Luminex instrument uses lasers to excite the beads, thereby identifying the bead region and its corresponding assigned analyte. The intensity of the PE-derived signal, which is directly proportional to the amount of bound analyte, was measured by another laser. Multiple measurements of the mean fluorescence intensity (MFI) were taken at each bead region to ensure robust detection. Cytokine concentrations were determined relative to a standard curve that plotted MFI against protein concentration.

### 2.11. Analysis of Serum Proteins Using ELISA

PD-1, PD-L1, and HO-1 concentrations in the serum were quantified using enzyme-linked immunosorbent assay (ELISA) kits (Abcam, Cambridge, MA, USA) for PD-1 (ab252360), PD-L1 (ab277712) and HO-1 (ab229429), as recommended by the manufacturer [44,45,46]. The serum concentrations of PD-1, PD-L1, and HO-1 were derived by evaluating the optical density of samples, followed by interpolation with standard curves.

### 2.12. Statistical Analysis

Statistical analyses were performed using SPSS (version 22.0; IBM Corp., Armonk, NY, USA) and GraphPad Prism software (version 9.5.1; GraphPad Software Inc., La Jolla, CA, USA). Since most variables were not normally distributed, the Mann–Whitney and Wilcoxon tests were applied for comparison of independent and dependent variables, respectively. All quantitative results were presented as the median with interquartile range, unless stated otherwise. Relative differences in the transcription level of the aforementioned proteins were determined by normalising to glyceraldehyde-3-phosphate dehydrogenase (GAPDH) using the 2^−ΔΔCT^ method [47]. The activity of PBMCs was assessed by computing the ratio between samples in wells with activators and the negative controls, employing either fluorometric or spectrophotometric assays. Resazurin metabolism in inactivated PBMCs was assessed via a fluorometric assay and subsequently normalised against the control group. The level of significance was assumed to be *p* < 0.05.

## 3. Results

### 3.1. Participant Characteristics

The median age of participants was 58 (14) years. The OC patients and controls did not differ in terms of age and BMI. The distribution of OC histological types was consistent with that observed in the general population [48]. Clinicopathological data for the OC patients and controls are shown in Table 1.

### 3.2. Immune-Related Protein Expression Is Downregulated in the PBMCs of OC Patients and Further Dysregulated Postoperatively

We conducted a comprehensive analysis of the expression of cytokines and immune-regulating proteins in PBMCs, focusing primarily on the mRNA level. As shown in Figure 2, significant decreases in the expression of almost all of the studied proteins were observed in the OC group prior to surgery compared to healthy controls, with the most pronounced being IL-6. The expression of IL-10 was not significantly different.

The OC group also showed changes in the interleukin expression in PBMCs during the early post-surgical period (Figure 3). Changes in interleukin mRNA expression were particularly pronounced on days 1 and 3 after surgery, mainly showing downregulation. Further to its unaltered expression in OC patients compared to healthy controls, IL-10 was the only interleukin that showed increased mRNA expression on the first day after surgical treatment. However, the expression of all interleukins in PBMCs showed a tendency to return to their pre-surgical state by day 5.

The changes in mRNA expression of other immune-related proteins in PBMCs varied. The decreased expression of PD-L1 was most pronounced on day 1 post-surgery. The expression of HO-1 and PD-1 decreased consistently throughout the observation period, reaching their lowest level on day 5 post-surgery. However, the decrease was only significant for HO-1 (Figure 4A). In parallel, the protein level of HO-1 in PBMCs peaked on day 1 after surgical treatment, followed by a return to the presurgical state (Figure 4B). All original immunoblots are included in Appendix A.

### 3.3. Surgical Treatment of OC Patients Affects PBMC Activity in the Early Post-Operative Period

No significant differences were detected in the metabolic and functional activities of PBMCs between OC patients before surgery and healthy participants. Nevertheless, a slight decline in PBMC activity could be seen in most of the observations (Figure 5). A significant decrease in metabolic activity was observed after surgery, yet this change was not evident in activated PBMCs. Additionally, trends for increased activity on post-operative day 5 were observed. A marked shift in the functional activity was also observed, especially on day 1 post-surgery. This was characterised by a significant increase in relative ROS production and a significant reduction in the relative phagocytic activity. However, by day 5 post-surgery, these changes had reverted to preoperative levels.

### 3.4. Surgical Treatment Alters the Serum Concentrations of Immune-Related Proteins in OC Patients

Finally, we evaluated the concentrations of immune-related proteins in the serum of participants. No significant differences in interleukin concentrations were observed between the controls and OC patients, even after the surgical treatment of OC patients (Figure 6A). The only exception was IL-6, which showed a marked increase on the first day after surgery, but then returned to baseline levels at day 5 post-surgery. Additionally, OC patients showed a trend for lower serum levels of the free PD-1 receptor compared to controls (Figure 6B). The HO-1 concentration was comparable between OC patients and controls. However, a consistent increase in the HO-1 concentration was observed during the post-operative period in OC patients, reaching a peak (possibly not final) on day 5 (Figure 6B).

## 4. Discussion

### 4.1. Suppression of PBMCs in OC Patients Compared to Healthy Controls

OC is not immunogenic in origin. However, the immune response is a crucial factor in OC progression and exerts a significant local impact [6,18], as well as affecting the systemic immune response in OC patients [18,49]. PBMCs are a source of anticancer immunity [6,16] and are recognised for their significance in cancer formation and personalised treatment [17,28]. The current study provides compelling evidence for a distortion in the immune status of OC patients in the early post-operative period. Notably, our findings indicate that the mRNA expression of interleukins and various immune-related proteins is downregulated in the PBMCs of OC patients before surgery. This phenomenon was consistent across the investigated cytokines and other immune-related proteins, irrespective of their immunosuppressive or immunostimulatory nature [27]. Although our results on differences in the metabolic and functional activity of PBMCs between OC and control groups did not reach statistical significance, previous research has shown reduced PBMC activity in OC patients [50,51,52]. In conjunction with results from the literature, our findings suggest that PBMCs are suppressed in OC patients.

The current study also investigated the serum levels of serum interleukins and immune-related proteins in OC patients. Although these can originate from various sources, including PBMCs [27], serum levels provide a more comprehensive view of the overall state of the body’s immune system. The concentration of serum cytokines in OC patients before surgery was not significantly different to that of the controls. However, a trend for lower IL-1β and IL-4 levels was observed, together with higher IL-6 and IL-10 levels. Some of these findings concur with the findings of other investigators [19,20,53], although the results from different studies vary. This may be explained by the fact that many factors influence cytokine concentrations, including the stage and histology of OC [54].

### 4.2. Post-Operative Changes in Interleukin Levels in the Serum and PBMCs of OC Patients

In a novel approach, we investigated PBMCs in the context of surgical trauma in OC. To comprehensively analyse the immune response in our study, we selected two proinflammatory cytokines (IL-1β and IL-6) and two anti-inflammatory cytokines (IL-4 and IL-10). These all play pivotal roles in OC progression [20,27,55,56,57]. Despite their known stimulatory or suppressive functions in the immune response, the properties of these cytokines may impact tumour progression. Notably, there is more compelling evidence for a pro-tumourigenic role of IL-6 [55,58,59] compared to IL-1β, IL-4, and IL-10, for which conflicting data exist [6,57,60]. Our results indicate that during the early post-operative period, the mRNA expression of IL-1β, IL-4, and IL-6 in PBMCs was significantly downregulated, while that of IL-10 was upregulated. Aside from IL-6, we observed consistent serum interleukin levels throughout the study period. Moreover, no significant correlations were found between serum interleukin levels and their mRNA expression level in PBMCs. Given the current ambiguity concerning the effect of these interleukins on cancer progression, it is difficult to establish a direct link between the observed changes in interleukin levels and their potential effects on OC. It is perhaps more prudent to speculate that PBMCs undergo some degree of overall suppression during the early post-operative period. However, we did observe a distinct increase in the serum IL-6 level during the early post-operative phase. This was expected, given the patients had undergone major surgery [61,62]. According to the available data, this surge in the IL-6 concentration creates favourable conditions for cancer progression [55,58,59].

### 4.3. Post-Operative Changes in PBMC and Serum Levels of HO-1, PD-1, and PD-L1 in OC Patients

Many mechanisms are known to be involved in tumour progression and in the development of chemoresistance [63]. PD-1/PD-L1 is a well-known immune checkpoint that participates in the mechanism that allows tumours to evade the immune system [29]. The expression of PD-1/PD-L1 on tumour and immune cells has been shown to have prognostic significance [28,64]. Furthermore, researchers are actively exploring PD-1/PD-L1 as a target for immune checkpoint blockade in the treatment of OC. Despite some promising results, these treatment methods have so far shown only limited efficacy [65,66]. This limitation can be attributed to a poor understanding of the mechanisms of these therapeutic agents, as well as to challenges in selecting responsive patients [29,65]. Efforts are underway to collect more data and overcome these obstacles [67], but additional fundamental knowledge regarding this immune checkpoint is still needed, especially for post-operative patients. In the current study, we observed a post-operative decrease in PD-L1 mRNA expression and a trend for increased PD-1 serum concentration after the low preoperative levels compared to the controls. The former may indicate that PBMCs shift towards increased immune tolerance during the early post-operative stage.

The heat shock protein HO-1 is an intracellular enzyme with antioxidant, anti-apoptotic, and cytoprotective properties. HO-1 has emerged as a critical factor associated with tumour progression in OC and other cancer types. It has been identified as a promoter of cancer cell proliferation, invasion, and migration, potentially fostering metastasis, although its precise mechanism of action remains uncertain [26,31]. HO-1 expression has been associated with cancer aggressiveness and poor clinical outcomes, making it a potential prognostic marker for OC patients [68]. Our findings suggest that during the early post-operative phase, OC patients may experience additional favourable conditions for tumour progression due to elevated post-operative serum levels of HO-1. This is despite a concurrent decrease in HO-1 mRNA expression within PBMCs.

### 4.4. Surgery Reduces the Activity of PBMCs in OC Patients

The current findings did not demonstrate a statistically significant reduction in the cytotoxic activity of immune cells in pre-operative OC patients. Nevertheless, previous studies support the notion of immune cell dysfunction in OC patients [51,52]. A particularly noteworthy result from the current work was the significant decline in PBMC phagocytic activity on day 1 after surgery in OC patients. Moreover, there was a notable decrease in baseline PBMC viability post-surgery, potentially impacting the results of PBMC function assays. However, following PBMC activation, metabolic activity rebounded. This suggests an independent decrease in PBMC functional activity post-surgery, despite the compromised viability. Increased ROS production was also observed during the same period. Although increased ROS production suggests heightened PBMC activity [69], high ROS levels can potentially promote OC progression [70,71], making this change less advantageous. A trend for the reversal of the day 1 post-surgical decrease in phagocytic activity was observed on day 5, indicating that the major changes in PBMC activity are likely to occur during the initial post-operative phase. It has also been reported that following optimal cytoreduction, peripheral immune cell activity in OC patients continues to normalise, eventually equalling that of healthy patients [72].

While there are additional data demonstrating immune suppression in OC patients post-surgery [73,74], it is essential to acknowledge that the observed changes might not be specific to the selected population. The existing data indicate that immune suppression following surgery extends beyond ovarian cancer, affecting various cancer types and even non-cancer populations [15,75]. Moreover, the extent of surgery and the type of anaesthesia employed significantly impact these outcomes [15,74]. Considering the extensive surgical interventions required for OC patients, investigating the post-surgery immune response becomes even more imperative and relevant.

### 4.5. Final Considerations Regarding the Post-Operative Immune Response in OC Patients

Given the significance of the immune response in OC progression [6], fluctuations in immune response dynamics could be critical for the disease course. Radical surgical treatment of OC results in positive impacts on the subsequent immune response [25] and greatly improves patient prognosis [4]. However, it may also have negative effects on metastasis formation [11]. Direct evidence linking the distortion of post-operative immune responses to metastasis formation in OC patients is lacking. By extrapolating results from other cancer types with regard to post-operative metastasis formation [9,11,12], as well as the evidence for immune suppression after surgery in OC patients [23,24,25], the current findings lead us to speculate that post-operative metastasis formation in OC patients could be a real phenomenon.

Our data revealed a significant suppression of the PBMC function in OC patients during the initial post-operative days, as well as dysregulated expression of immune-related proteins. We consider this imbalance in PBMC activity as a potential vulnerability that could facilitate the progression of OC. Therefore, further investigation of the post-operative immune response is crucial, with a specific focus on PBMCs. Given the challenges in OC treatment, such as chemoresistance [63] and frequent disease relapse [6], we propose that the early post-operative period presents a valuable opportunity to enhance concurrent treatment strategies. These could include systemic intraoperative chemotherapy or hyperthermic intraperitoneal chemotherapy [7,76]. The establishment of patient selection criteria based on PBMC activity in the early post-operative phase could also play a pivotal role in developing new treatments and refining existing treatment modalities for OC patients.

## 5. Conclusions

In conclusion, this study has provided further insights into early post-operative immune changes in OC patients, revealing significant dysregulation in PBMC activity and immune-related protein expression. Surgical treatment exacerbates the pre-operative suppression of PBMC activity and causes changes in the mRNA expression levels of IL-1β, IL-4, IL-6, IL-10, HO-1, PD-1, and PD-L1 in PBMCs. While the interpretation of this dysregulation can be complex, notable fluctuations in serum IL-6 and HO-1 levels, together with suppressed phagocytosis by PBMCs, indicate a shift towards immune tolerance post-surgery. Although not all findings reached statistical significance, the overall trend indicates compromised PBMC activity in OC patients during the early post-operative period, with the most significant changes seen on the first day after surgery. These findings emphasise the need to assess the impact of surgery on the immune response during the early post-operative phase, as this may create a vulnerable window for OC to progress. Further refinement of patient selection strategies based on the post-operative activity of PBMCs could optimise treatment approaches and improve the long-term prognosis of OC patients.

## 6. Study Limitations

While this study provides valuable insights, it has some limitations. The relatively small sample size may limit the generalisability of the findings. However, given the novelty and scale of experiments conducted on post-surgical OC patients, sample size calculations were challenging. Additionally, although we evaluated changes in PBMC activity and immune-related protein expression, our study did not assess their impact on disease progression and clinical outcomes. Moreover, we did not include a control group consisting of individuals with a benign gynaecological pathology to examine the specific impact of OC to our results. Considering the extensive nature of surgery for OC patients, selecting suitable patients for an adequate control group is challenging.

## 7. Practical Recommendations

Future research with larger cohorts is necessary to validate the current findings and gain a better understanding of the clinical implications of immune dysregulation in OC patients.

## Figures and Tables

**Figure 1 cancers-16-00190-f001:**
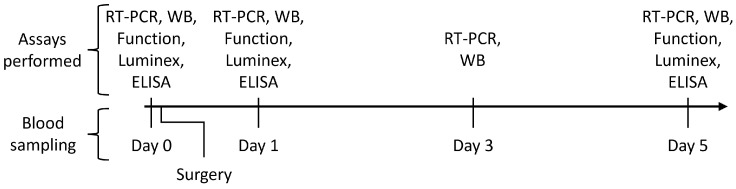
Blood sampling and assay schedule for the ovarian cancer (OC) patients. The lower section of the timeline shows the four specific time points for blood collection. The upper section of the figure outlines the assays conducted at each blood sampling: RT-PCR, real-time polymerase chain reaction; WB, Western blot assay; function, assessment of peripheral blood mononuclear cells (PBMC) metabolic and functional activity; Luminex assay; ELISA assay.

**Figure 2 cancers-16-00190-f002:**
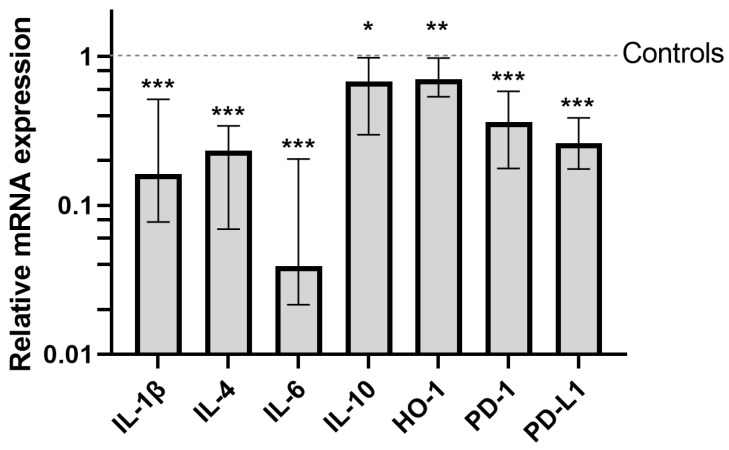
The mRNA expression of interleukins, HO-1, PD-1, and PD-L1 was significantly downregulated in the PBMCs of OC patients before surgery, with the exception of IL-10. The histogram shows the relative expression of mRNA compared to healthy controls, with the values normalised to one. * *p* > 0.05; ** *p* = 0.02; *** *p* ≤ 0.001.

**Figure 3 cancers-16-00190-f003:**
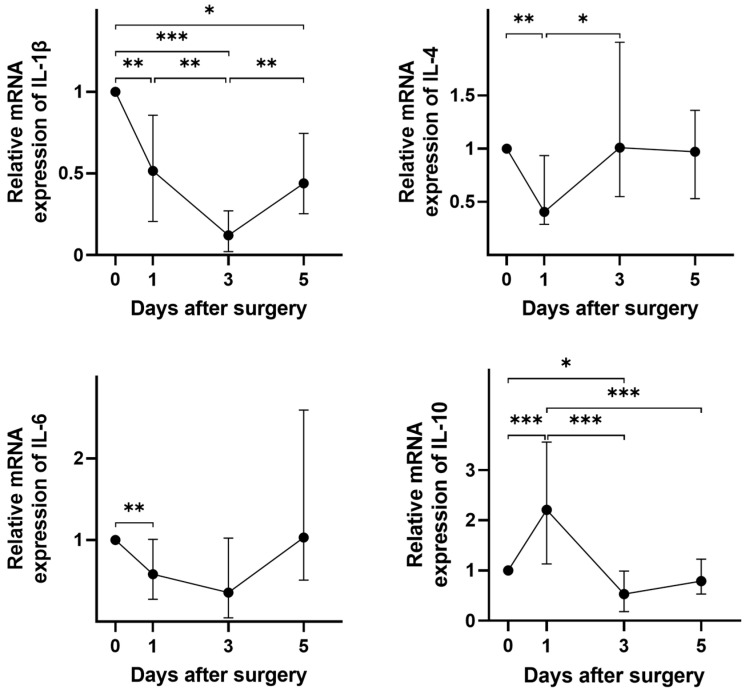
Surgical treatment alters the expression of interleukin mRNAs in the PBMCs of OC patients. The relative expression levels of different interleukin mRNAs are shown at days 1, 3, and 5 post-surgery compared to the pre-surgical baseline. All *p*-values comparing the different days were >0.05 unless otherwise specified. * *p* < 0.05; ** *p* ≤ 0.01; *** *p* ≤ 0.001.

**Figure 4 cancers-16-00190-f004:**
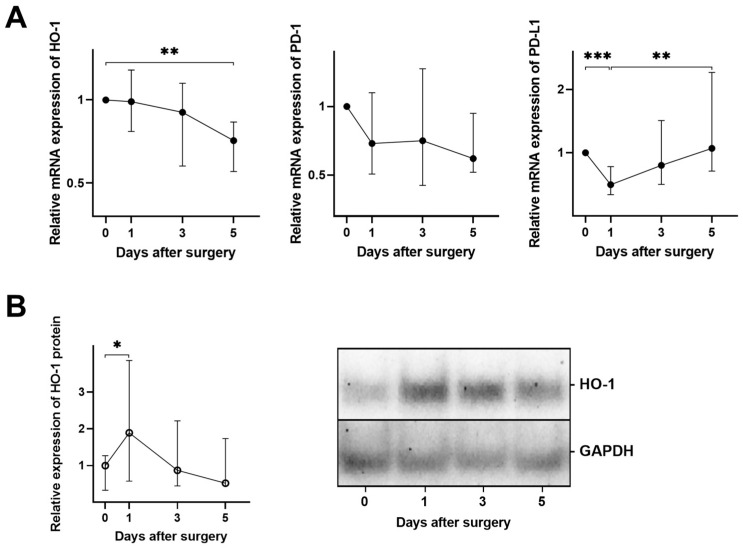
Surgical treatment causes dysregulation of HO1, PD-1, and PD-L1 expression in PBMCs from OC patients. (**A**) Relative expression of HO-1, PD-1, and PD-L1 mRNAs at days 1, 3, and 5 after surgical treatment compared to the pre-surgical baseline. (**B**) Changes in the HO-1 protein level in PBMCs (left), as determined by densitometric scanning of immunoblots and normalised to GAPDH (representative immunoblot on the right). All *p*-values comparing samples from different days were >0.05 unless otherwise specified. * *p* < 0.05; ** *p* ≤ 0.01; *** *p* ≤ 0.001.

**Figure 5 cancers-16-00190-f005:**
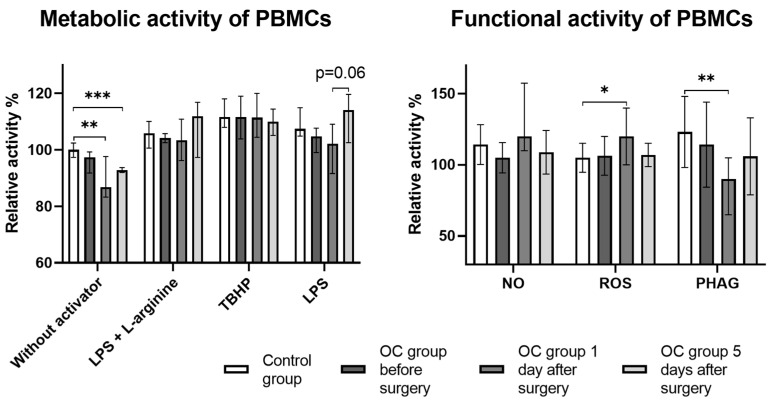
PBMC activity in healthy controls and OC patients before and after surgical treatment. Functional activity (**right panel**) encompassed the evaluation of nitric oxide (NO) and reactive oxygen species (ROS) production, alongside measuring phagocytic activity (PHAG). Metabolic activity (**left pane**l) represents PBMCs viability assessed via the resazurin metabolism assay (AlamarBlue) without an activator, and subsequently repeated with the same activators used in the functional activity assays (LPS, lipopolysaccharide; L-arginine; TBHP, tert-butyl hydroperoxide). The relative activity shown was computed by comparing fluorometric or spectrophotometric assay results between observations with and without activators. Resazurin metabolism without activators was assessed using a fluorometric assay and then normalised to the control group. All *p*-values between groups are >0.05 unless stated otherwise. * *p* = 0.02; ** *p* ≤ 0.01; *** *p* = 0.001.

**Figure 6 cancers-16-00190-f006:**
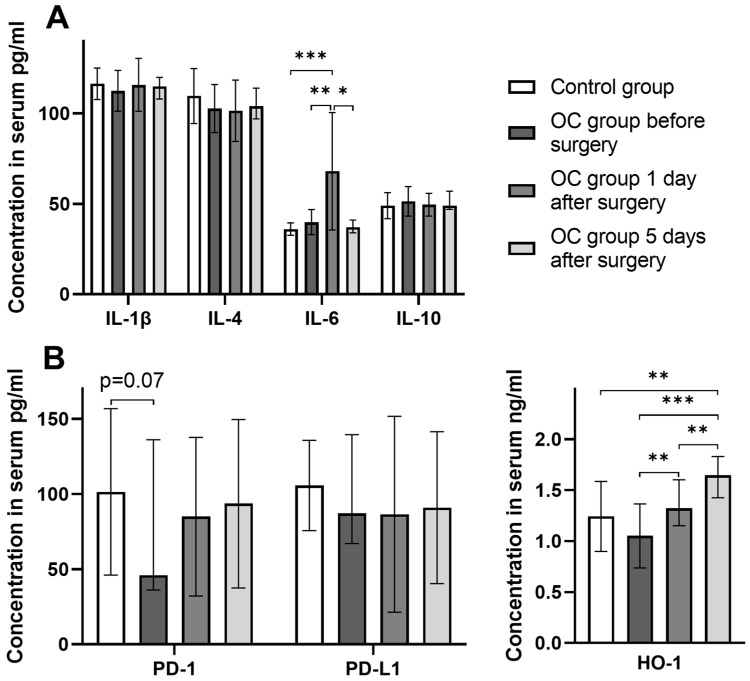
Changes in serum concentrations of interleukins, HO-1, PD-1, and PD-L1 in pre- and post-operative OC patients. (**A**) Serum levels of interleukins evaluated using Luminex assay. (**B**) Serum levels of PD-1, PD-L1, and HO-1 evaluated by the ELISA method. All *p*-values between groups are >0.05 unless stated otherwise. * *p* < 0.05; ** *p* ≤ 0.01; *** *p* ≤ 0.001.

**Table 1 cancers-16-00190-t001:** Clinicopathological characteristics of the ovarian cancer (OC) and control groups.

Characteristic	OC Group (*n* = 23)	Control Group (*n* = 20)	*p*-Value
Age (years) *	58 (14)	60.5 (10)	0.48
Body mass index (kg/m^2^) *	24 (7.4)	24.7 (6.6)	0.97
Stage of OC **			
IIIA	4 (17.4)	NA	NA
IIIB	4 (17.4)	NA	NA
IIIC	7 (30.4)	NA	NA
IVA	1 (4.4)	NA	NA
IVB	7 (30.4)	NA	NA
Histological type of OC **			
Low-grade serous carcinoma	1 (4.4)	NA	NA
High-grade serous carcinoma	15 (65.1)	NA	NA
Endometrioid carcinoma	2 (8.7)	NA	NA
Clear cell carcinoma	1 (4.4)	NA	NA
Mucinous carcinoma	1 (4.4)	NA	NA
Serous endometrioid carcinoma	3 (13)	NA	NA

* median with interquartile range; ** number of cases with percentage. The stage of OC is presented according to FIGO [32].

## Data Availability

The data presented in this study are available on request from the corresponding author.

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
