# Peer review of "Dysregulation of Peripheral Blood Mononuclear Cells and Immune-Related Proteins during the Early Post-Operative Immune Response in Ovarian Cancer Patients"

_cancers, 2023, doi:10.3390/cancers16010190_

Round 1
Reviewer 1 Report
Comments and Suggestions for Authors
1.Did patients receive neoadjuvant chemotherapy before surgery?
2.Was the viability of PBMCs determined after their isolation from peripheral blood? And did this indicator differ before surgery and at different times after surgery in patients with ovarian cancer?
3.Figure 2 shows the relative values of cytokine expression in ovarian cancer patients compared to the control group in which period? Before or after surgery? Results need clarification
4Did you compare the level of cytokine expression between groups of patients of stage IIIA, IIIB and IIIC?
5The conducted research on immune indicators in patients with ovarian cancer in the preoperative period and in the early days after surgery cannot to a certain extent be a prognostic marker of the progression of neoplasms, it would be worthwhile to pay attention to the determination of the expression of these cytokines in more distant periods of the postoperative period. It is also important to consider the scheme of adjuvant chemotherapy or hormone therapy, which can be a powerful factor in suppressing the immune system in patients with ovarian cancer.
Author Response
Dear Sir or Madam,
We would like to express our sincere gratitude for the invaluable insights and thoughtful observations provided during the review of our manuscript. Your feedback has been instrumental in refining the quality of our work, and we truly appreciate the time and effort you dedicated to offering such constructive comments. We have meticulously addressed each of the questions and notions raised, channelling our utmost dedication and commitment to enhancing the manuscript's coherence, accuracy, and overall strength.
This response includes a detailed point-by-point address of the questions raised, specifically referencing the manuscript's lines for clarity. It's important to note that while we focused on the highlighted sections addressing the questions raised by you, additional alterations have been made throughout the manuscript to accommodate other reviewers' inquiries.
Point-by-point response to Comments and Suggestions for Authors
Comment 1: Did patients receive neoadjuvant chemotherapy before surgery?
Response 1: Ovarian cancer patients with neoadjuvant chemotherapy were not included in our study. We specifically chose this patient group to eliminate any potential impact of chemotherapy on the patients immune systems and subsequently our study findings. This information has been incorporated into the manuscript (page 3, paragraph 2.1., line 101).
Comment 2: Was the viability of PBMCs determined after their isolation from peripheral blood? And did this indicator differ before surgery and at different times after surgery in patients with ovarian cancer????
Response 2: We've made clarifications in the Methods and Results sections to indicate that metabolic activity represents PBMCs viability. Additionally, we've integrated data showcasing resazurin metabolism without activators into Figure 5, offering insights into baseline viability and its alterations post-surgery. These new findings required additional commentary and discussion, now included in the manuscript (page 5, paragraph 2.6., lines 195-196 and 202-204; page 10, paragraph 3.3., lines 341-343; page 11, paragraph 3.3, lines 351-354 and 356-357; page 14, paragraph 4.4., lines 453-457).
Comment 3: Figure 2 shows the relative values of cytokine expression in ovarian cancer patients compared to the control group in which period? Before or after surgery? Results need clarification.
Response 3: Figure 2 illustrates the difference of cytokine production in PBMCs between healthy controls and ovarian cancer patients before surgery. Post-surgical cytokine expressions in OC patients are depicted in subsequent figures. This clarification has been included in the manuscript for better coherence (pages 7-8, paragraph 3.2, lines 304 and 308; page 12, paragraph 4.1., line 384). Additionally, we've made edits to the figure by refining the “star” labelling and providing an explanation for the line denoting normalised values of the controls, addressing feedback from another reviewer (page 8, paragraph 3.2, line 306).
Comment 4: Did you compare the level of cytokine expression between groups of patients of stage IIIA, IIIB and IIIC?
Response 4: We further explored cytokine expression in PBMCs before and after surgery among OC patients across various stages. However, no significant differences were identified. This outcome is anticipated considering the relatively limited sample size, particularly when subdivided for comparison among different stages of OC patients. Despite the significance of this aspect, due to the aforementioned constraints, we question the need of publishing these data.
Comment 5: The conducted research on immune indicators in patients with ovarian cancer in the preoperative period and in the early days after surgery cannot to a certain extent be a prognostic marker of the progression of neoplasms, it would be worthwhile to pay attention to the determination of the expression of these cytokines in more distant periods of the postoperative period. It is also important to consider the scheme of adjuvant chemotherapy or hormone therapy, which can be a powerful factor in suppressing the immune system in patients with ovarian cancer.
Response 5: We greatly appreciate your valuable insights. We concur with your perspective; our study's brief investigation period (5 days post-surgery for each ovarian cancer patient) precludes the examination of long-term patient outcomes or prognosis, which falls beyond the scope of our research. Nevertheless, we firmly believe that exploring early postsurgical immunity in ovarian cancer patients is a crucial area that warrants attention, given its potential influence on ovarian cancer progression.
Undoubtedly, chemotherapy significantly impacts immune response. In our study, we deliberately focused on patients who did not undergo neoadjuvant chemotherapy to eliminate potential confounding impacts, as our primary aim was to investigate post-surgical immune imbalances.
Reviewer 2 Report
Comments and Suggestions for Authors
Major comment:
There is no pre- and post-surgery samples from patients with non-malignant gynaecological diseases.
How do the authors know the changes they observed (IL-1B mRNA (Figure 3A), IL-4 mRNA (Figure 3A), IL-6 mRNA (Figure 3A); HO-1 mRNA (Figure 4A), PD-1 mRNA (Figure 4A); IL-6 protein concentration (Figure 6A)) are not due to the wound healing process post-surgery?
Minor comment:
Please emphasize using the PBMC of OC patients prior to surgery in Figure 2.
Author Response
Dear Sir or Madam,
We would like to express our sincere gratitude for the invaluable insights and thoughtful observations provided during the review of our manuscript. Your feedback has been instrumental in refining the quality of our work, and we truly appreciate the time and effort you dedicated to offering such constructive comments.
This response includes a detailed point-by-point address of the questions raised, specifically referencing the manuscript's lines for clarity. It's important to note that while we focused on the highlighted sections addressing the questions raised by you, additional alterations have been made throughout the manuscript to accommodate other reviewers' inquiries.
Point-by-point response to Comments and Suggestions for Authors
Comment 1: There is no pre- and post-surgery samples from patients with non-malignant gynaecological diseases. How do the authors know the changes they observed (IL-1B mRNA (Figure 3A), IL-4 mRNA (Figure 3A), IL-6 mRNA (Figure 3A); HO-1 mRNA (Figure 4A), PD-1 mRNA (Figure 4A); IL-6 protein concentration (Figure 6A)) are not due to the wound healing process post-surgery?
Response 1: Thank you for your valuable insight. Despite existing scant data, similar to our study, it seems that the observed changes aren't specific to our study population. Immune changes are evident not only in cancer patients but also in non-cancer patients and are notably influenced by factors like the type of surgery or anaesthesia used. Minimally invasive surgeries and preference to local anaesthesia, appear to exert a lesser impact on the immune response. We've integrated these considerations into the manuscript (page 14, paragraph 4.4., lines 465-472; page 15, paragraph 6, lines 518-521).
We also deliberated on the prospect of incorporating an additional study group comprising patients with benign gynaecological pathologies undergoing surgical treatment. However, we encountered a challenge in finding benign pathologies necessitating similar surgical procedures in terms of length and extent of the surgery (total hysterectomy, omentectomy, peritoneal resection). Consequently, obtaining a suitable quality control group appeared infeasible. Alternatively, introducing a control group of benign pathologies with smaller surgeries, involving different anaesthesia methods, would likely introduce additional disparities, complicating result interpretation.
Yet, we cannot disregard the observed immune imbalances in post-surgical OC patients, even if they could be attributed to the healing process and other factors. Regardless of their origins, we think that these fluctuations might substantially impact the progression of OC. Although our research didn't specifically explore the impact of ovarian cancer itself on post-surgical immune imbalances, this remains a crucial area deserving further investigation.
Comment 2: Please emphasize using the PBMC of OC patients prior to surgery in Figure 2.
Response 2: This clarification has been included in the manuscript for better coherence (pages 7-8, paragraph 3.2, lines 304 and 308; page 12, paragraph 4.1., line 384). Additionally, we've made edits to the figure by refining the “star” labelling and providing an explanation for the line denoting normalised values of the controls, addressing feedback from another reviewer (page 8, paragraph 3.2, line 306).
Reviewer 3 Report
Comments and Suggestions for Authors
In the manuscript entitled “Dysregulation of PBMCs and Immune-Related Proteins During the Early Post-Operative Immune Response in Ovarian Cancer Patients”, Ulevicius and colleagues investigated the potential alterations in PBMC activity, concentration of interleukins, and other immune-related proteins in surgically treated Ovarian Cancer patients. They provided insights into the dysregulation of the immune response during the early post-operative phase. This work offered interesting findings for potential treatment strategies to ovarian cancer. The opinions will certainly be of importance to the field.
However, I have listed below a series of suggestions which would further improve the quality of the manuscript. The questions and concerns must be addressed before the publication of this paper.
1. Why were these cytokines and immune regulating proteins selected? It will be appreciated if the authors could touch base upon them either in the introduction or results before the discussion.
2. In Figure 1, what are the reasons for the missing assays, such as Function, Luminex and Elisa for day 3? Please explain in 2.2 if possible.
3. The Figure 2 is confusing. Why did the authors only label one star, while the other groups look more prominently decreased? When was this set of mRNAs measured, day 1, 3, 5?
4. The explanation for Figure 5 is not enough. Why were the NO, ROS production, and PHAG selected?
What’s the difference for measuring the metabolic and functional activities of PBMCs? And why there is no day 3? Please provide more details for this set of assays.
5. In the discussion, the authors should also consider whether the immune response changes are specific to OC surgery or the same to any other surgery post-operative phase. Are there any other similar or related studies for the 1st day after surgery?
6. The authors should pay more attention to the details. The problems include but are not limited to: In Figure 6A, IL-1b should be IL-1β?
Please cite the Figure S1 in appropriate places.
Author Response
Dear Sir or Madam,
We would like to express our sincere gratitude for the invaluable insights and thoughtful observations provided during the review of our manuscript. Your feedback has been instrumental in refining the quality of our work, and we truly appreciate the time and effort you dedicated to offering such constructive comments. We have meticulously addressed each of the questions and notions raised, channelling our utmost dedication and commitment to enhancing the manuscript's coherence, accuracy, and overall strength.
This response includes a detailed point-by-point address of the questions raised, specifically referencing the manuscript's lines for clarity. It's important to note that while we focused on the highlighted sections addressing the questions raised by you, additional alterations have been made throughout the manuscript to accommodate other reviewers' inquiries.
Point-by-point response to Comments and Suggestions for Authors
Comment 1: Why were these cytokines and immune regulating proteins selected? It will be appreciated if the authors could touch base upon them either in the introduction or results before the discussion.
Response 1: Upon your request, we have included an introduction to these cytokines and immune-regulating proteins in the Introduction section (page 2, paragraph 1., lines 75-88). Furthermore, they are elaborated upon in the Discussion section (page 13, paragraph 4.2., lines 402-408; page 13, paragraph 4.3., lines 424-434 and 438-444).
Comment 2: In Figure 1, what are the reasons for the missing assays, such as Function, Luminex and Elisa for day 3? Please explain in 2.2 if possible.
Response 2: After analysing the pilot results, we noted a consistent change in data between days 1 and 5 following the surgery. As a result, we opted to omit specific assays from the day 3 protocol. This update has been included in the manuscript to enhance clarity (page 3, paragraph 2.2., lines 115-117). Additionally, this decision was influenced by resource limitations for conducting the study.
Comment 3: The Figure 2 is confusing. Why did the authors only label one star, while the other groups look more prominently decreased? When was this set of mRNAs measured, day 1, 3, 5?
Response 3: Figure 2 illustrates the difference of cytokine production in PBMCs between healthy controls and ovarian cancer patients before surgery. Post-surgical cytokine expressions in OC patients are depicted in subsequent figures. This clarification has been included in the manuscript for better coherence (pages 7-8, paragraph 3.2, lines 304 and 308; page 12, paragraph 4.1., line 384). We've also made edits to the figure by refining the “star” labelling and providing an explanation for the line denoting normalised values of the controls (page 8, paragraph 3.2., line 306).
Comment 4.1: The explanation for Figure 5 is not enough. Why were the NO, ROS production, and PHAG selected?
Response 4.1: Nitric oxide (NO) and reactive oxygen species (ROS) production, along with phagocytosis, are key indicators of immune cell function in the context of anti-cancer immunity. These processes are crucial components of the immune response against cancer cells, contributing to tumour cell destruction and anti-tumor immune activation. Further details explaining the rationale for selecting these markers have been provided in the respective paragraphs within the Methods section (page 5, paragraph 2.7., lines 207-211; page 5, paragraph 2.8., lines 224-227; page 6, paragraph 2.9., lines 238-241).
Comment 4.2: What’s the difference for measuring the metabolic and functional activities of PBMCs? And why there is no day 3? Please provide more details for this set of assays.
Response 4.2: We've clarified in the Methods and Results sections that metabolic activity denotes PBMCs viability, while functional activity pertains to specific immune mechanisms of PBMCs. Additionally, we've integrated data showcasing resazurin metabolism without activators into Figure 5, offering insights into baseline viability and its alterations post-surgery. These new findings required additional commentary and discussion, now included in the manuscript (page 5, paragraph 2.6., lines 195-196 and 202-204; page 10, paragraph 3.3., lines 341-343; page 11, paragraph 3.3., lines 351-354 and 356-357; page 14, paragraph 4.4., lines 453-457).
As aforementioned, upon reviewing the pilot results, we observed consistent data changes from days 1 to 5 post-surgery. Consequently, we excluded certain assays from the day 3 protocol. Resource constraints also factored into this decision (page 3, paragraph 2.2., lines 115-117).
Comment 5: In the discussion, the authors should also consider whether the immune response changes are specific to OC surgery or the same to any other surgery post-operative phase. Are there any other similar or related studies for the 1st day after surgery?
Response 5: Thank you for your valuable insight. Despite existing scant data, similar to our study, it seems that the observed changes aren't specific to our study population. Immune suppression is evident not only in cancer patients but also in non-cancer patients and is notably influenced by factors like the type of surgery or anaesthesia used. Minimally invasive surgeries and preference to local anaesthesia, appear to exert a lesser impact on the immune response. We've integrated these considerations into the manuscript (page 14, paragraph 4.4., lines 465-472; page 15, paragraph 6., lines 518-521).
Comment 6.1: The authors should pay more attention to the details. The problems include but are not limited to: In Figure 6A, IL-1b should be IL-1β?
Response 6.1: Thank you for pointing that out. We have made the necessary corrections to the figure (page 12, paragraph 3.4., line 370).
Comment 6.2: Please cite the Figure S1 in appropriate places.
Response 6.2: Thank you for bringing this to our attention. We have included a reference in the manuscript (page 9, paragraph 3.2., line 329).
Round 2
Reviewer 3 Report
Comments and Suggestions for Authors
The authors have promptly and largely responded and addressed the major concerns. The manuscript quality was significantly improved.
Here, I still have some suggestions for further revising the manuscript. When solved, I would suggest the publication of this paper.
In Figure 2, the significant stars are not consistent with other figures, actually reversed. This is confusing. Please consider revising.
For the new data of metabolic activity without activator in Figure 5, the authors should also compare groups after surgery with the group before surgery (just like Figure 6A IL-6). Please consider analyzing and adding the stats.
Author Response
I wish to express my appreciation to the reviewer for insightful comments and thorough review of our manuscript. Following the previous round of revisions, we have meticulously addressed all raised concerns in line with the journal's guidelines. All modifications have been highlighted and referenced for ease of review by the editors and reviewers.
Comment 1. In Figure 2, the significant stars are not consistent with other figures, actually reversed. This is confusing. Please consider revising.
Response 1. Thank you for pointing this out. We've rectified the issue in Figure 2 as well as in the corresponding description below the figure, as detailed in the revised manuscript (page 8, paragraph 3.2., lines 306 and 310).
Comment 2. For the new data of metabolic activity without activator in Figure 5, the authors should also compare groups after surgery with the group before surgery (just like Figure 6A IL-6). Please consider analyzing and adding the stats.
Response 2. Thank you for highlighting this point. We conducted a comparison of metabolic activity among all groups, including the pre- and post-surgery groups. However, no statistically significant differences, apart from those indicated in the graph, were observed. We have now explicitly stated in the manuscript, below the figure (page 11, paragraph 3.3., lines 357-358) that "all p-values between groups are >0.05 unless stated otherwise".
Best regards,
Jonas Ulevicius